# Neuroprotective Effect of 1,3-dipalmitoyl-2-oleoylglycerol Derived from Rice Bran Oil against Cerebral Ischemia-Reperfusion Injury in Rats

**DOI:** 10.3390/nu14071380

**Published:** 2022-03-25

**Authors:** Hong Kyu Lee, Ji Yeon Jang, Hwan-Su Yoo, Yeon Hee Seong

**Affiliations:** 1College of Veterinary Medicine, Chungbuk National University, Cheongju 28644, Korea; hklee83@cbnu.ac.kr (H.K.L.); jjy8401@hanmail.net (J.Y.J.); 2Institute for Stem Cell & Regenerative Medicine (ISCRM), Chungbuk National University, Cheongju 28644, Korea; 3College of Pharmacy, Chungbuk National University, Cheongju 28160, Korea; yoohs@cbnu.ac.kr

**Keywords:** 1,3-dipalmitoyl-2-oleoylglycerol, triacylglyceride (TAG), ischemic stroke, MCAO/reperfusion, neuroprotection

## Abstract

1,3-Dipalmitoyl-2-oleoylglycerol (POP) is a triacylglyceride found in oils from various natural sources, including palm kernels, sunflower seeds, and rice bran. In the current study, the neuroprotective effects and the specific mechanism of POP derived from rice bran oil were investigated for the first time using the middle cerebral artery occlusion/reperfusion (MCAO/R) model in rats. Orally administered POP at 1, 3, or 5 mg/kg (three times: 0.5 h before MCAO, after 1 h of MCAO, and after 1 h of reperfusion) markedly reduced the MCAO/R-induced infarct/edema volume and neurobehavioral deficits. Glutathione depletion and the oxidative degradation of lipids in the rat brain induced by MCAO/R were prevented by POP administration. The upregulation of phosphorylated p38 MAPKs, inflammatory factors (inducible nitric oxide synthase (i-NOS) and cyclooxygenase-2 (COX-2)), and pro-apoptotic proteins (B-cell lymphoma-2 (Bcl-2) associated X protein (Bax) and cleaved caspase-3) and the downregulation of the anti-apoptotic protein (Bcl-2) in the ischemic brain were significantly inhibited by POP administration. In addition, downregulation of phosphatidylinositol 3′-kinase (PI3K), phosphorylated protein kinase B (Akt), and phosphorylated cyclic (adenosine monophosphate) AMP responsive element-binding protein (CREB) expression in the ischemic brain was inhibited by POP administration. These results suggest that POP might exert neuroprotective effects by inhibition of p38 MAPK and activation of PI3K/Akt/CREB pathway, which is associated with anti-oxidant, anti-apoptotic, and anti-inflammatory action. From the above results, the present study provides evidence that POP might be effectively applied for the management of cerebral ischemia-related diseases.

## 1. Introduction

Stroke is one of the major causes of death and long-term disability in the elderly worldwide, with high social and economic burdens for management [1]. Ischemic stroke, the predominant type of all stroke cases, occurs when blood flow to the brain is severely interrupted [1,2]. Although recombinant tissue plasminogen activator (rTPA) has been approved for the treatment of ischemic stroke as a thrombolytic agent, the clinical effectiveness of rTPA is limited by a narrow therapeutic time window (~3 to 4 h after ischemia onset) and the possibility of a cerebral hemorrhage [3,4]. Moreover, reperfusion by thrombolysis aggravates secondary brain damage over time via numerous mechanisms, including excitotoxicity, oxidative stress, inflammation, and apoptosis, which are believed to be interrelated [5,6]. With the remarkable advances in neuroscience research over the last few decades, various strategies to counteract mechanisms involved in ischemia-reperfusion injury for stroke treatment have been tried, but the development of an effective treatment for ischemic stroke is still elusive [2]. Therefore, there is a need to develop neuroprotective agents that alleviate cerebral ischemia-reperfusion injury.

Mitogen-activated protein kinases (MAPKs), including extracellular signaling-regulating kinases, c-Jun *N*-terminal kinase, and the p38 pathway, are pleiotropic serine/threonine-protein kinases that convert diverse extracellular signals into various cellular responses, including differentiation, proliferation, and apoptosis [7,8]. As a member of MAPKs, the p38 MAPK pathway is known to be associated with neuronal apoptosis and survival after transient cerebral ischemia [9,10]. Activation of the p38 MAPK pathway mediates neuroinflammation through upregulation of inducible nitric oxide synthase (i-NOS)/cyclooxygenase-2 (COX-2) signaling and elevates reactive oxygen species (ROS) production, consequently resulting in expansion of the brain infarct area [11,12]. In addition, inhibition of the p38 MAPK pathway contributes to neurological recovery after cerebral ischemia-reperfusion insult by upregulating phosphatidylinositol 3′-kinase (PI3K)/protein kinase B (Akt) signaling [13,14,15]. Upregulation of PI3K/Akt signaling has been reported to inactivate pro-apoptotic factors such as caspase-3 and Bcl-2 associated X protein (Bax) in the brain [16]. Therefore, the p38 MAPK pathway may be a potential therapeutic target for ischemic stroke.

Triacylglyceride (TAG) is the major component of dietary fats and oils and plays pivotal roles as a nutrient and energy source in the body. TAG consists of a glycerol moiety and three fatty acids that are esterified at three distinct positions (stereospecifically numbered *sn*-1, *sn*-2, and *sn*-3) on the glycerol. For 1,3-dipalmitoyl-2-oleoylglycerol (POP), palmitic acid is located at the external *sn*-1 and *sn*-3 positions, and oleic acid is located at the internal *sn*-2 position. POP is one of the major triacylglycerol components found in various natural oils, including theobroma oil, palm oil, and rice bran oil (RBO) [17,18]. Although the cholesterol-lowering ability of POP has been reported, there are few other studies related to the physiological activity of POP [19]. Previously, we isolated POP from RBO and confirmed that POP derived from RBO possesses neuroprotective activity against *N*-methyl-D-aspartate-mediated excitotoxicity in cultured cortical neurons, suggesting that POP might be a potential agent for treating cerebral ischemia-reperfusion injury. In the current study, the neuroprotective effect and specific mechanisms of POP were identified for the first time using an in vivo middle cerebral artery occlusion/reperfusion (MCAO/R) model in rats.

## 2. Materials and Methods

### 2.1. Preparation of POP

POP was isolated from RBO. The RBO was extracted from rice bran (Ochang Rice Processing Center, Chungbuk, Korea) as described previously [20] and an active fraction of RBO was purified by silica gel column chromatography extracted with hexane-ethyl acetate (10:1, *v/v*), followed by Sephadex LH-20 column chromatography (Sigma-Aldrich, St. Louis, MO, USA) using chloroform-methanol (1:1, *v/v*). The active component POP was purified with a silica Sep-Pak cartridge extracted with hexane-ethyl acetate (10:1, *v/v*) and identified by the mass measurement and nuclear magnetic resonance (NMR) spectroscopic method. In brief, electron ionization-mass measurement and NMR spectra, including ^1^H NMR, ^13^C NMR, ^1^H-^1^H COSY (correlation spectroscopy), HMQC (Heteronuclear Multiple Quantum Coherence), and HMBC (Heteronuclear Multiple Bond Coherence) spectra confirmed the presence of one glycerol, two palmitoyl, and one oleoyl moieties. The presence of two oxygenated carbon peaks in glycerol moiety demonstrated that this compound should have the symmetrical structure of POP (Figure 1).

### 2.2. Chemicals and Reagents

A bicinchoninic acid (BCA) protein assay kit, 2,3,5-triphenyl tetrazolium chloride (TTC), 5,5′-dithiobis-2-nitrobenzoic acid were obtained from Sigma Chemical Co. (St. Louis, MO, USA). The Pro-prep protein extraction buffer was purchased from iNtRONBio. Inc. (Seongnam-Si, Korea). Thiobarbituric acid (TBA) was purchased from Tokyo Kasei Kogyo Co., Ltd. (Tokyo, Japan). Rabbit monoclonal antibodies against p-38 MAPK (p38; clone D13E1), phospho-p38 (p-p38; Thr180/Tyr182; clone D3F9), Akt (clone 11E7), phosphorylated protein kinase B (p-Akt) (Ser473; clone D9E), cyclic adenosine monophosphate (AMP) responsive element-binding protein (CREB; clone 48H2), and phosphorylated cyclic AMP responsive element-binding protein (p-CREB) (Ser133; clone 87G3), and rabbit polyclonal antibodies against B-cell lymphoma-2 (Bcl-2) were obtained from Cell Signaling Technology, Inc. (Danvers, MA, USA). Rabbit polyclonal antibodies against Bcl-2 associated X protein (Bax), cleaved caspase-3, procaspase-3, i-NOS, PI3K, and β-actin, goat polyclonal antibody against COX-2, and horseradish peroxidase (HRP)-conjugated anti-rabbit secondary antibody was obtained from Millipore, Inc. (Bedford, MA, USA). HRP-conjugated anti-goat secondary antibody was purchased from Assay Designs (Ann Arbor, MI, USA). All other chemicals were of the highest grade available.

### 2.3. Experimental Animals 

Male Sprague–Dawley (SD) rats for MCAO/R were purchased from Samtako, Inc. (Osan, Korea), and maintained in an environmentally controlled space (specific pathogen-free at 22 ± 2 °C with a relative humidity of 55 ± 5% and a 12-h light/dark cycle). This animal study was conducted in accordance with the regulations of the Institutional Animal Care and Use Committee of Chungbuk National University (CBNUA-924-16-02).

### 2.4. MCAO/R-Induced Transient Cerebral Ischemia in Rats and POP Treatment

Prior to surgery, male SD rats (280 g~300 g) were fasted overnight with free access to water. Transient focal cerebral ischemia was generated as previously described [21]. Briefly, rats were anesthetized with 2.0 to 2.5% (*v/v*) isoflurane in an airflow using a laboratory animal inhalation anesthesia system (VetEquip, Inc., Livermore, CA, USA). A monofilament nylon suture with a silicone-coated distal end was inserted into the internal carotid artery to induce transient MCAO. After 2 h of MCAO, the animal was re-anesthetized, and the monofilament suture was removed to induce reperfusion injury. Sham surgery animals were operated on with the same procedure, except that monofilament sutures were inserted. The rats were placed on a heating pad to maintain their body temperature in the normal range (36.5 ± 0.5 ℃), and their rectal temperatures were monitored during the MCAO/R. POP was dissolved in dimethylsulfoxide (100 mg/mL) and diluted further in saline. POP (1, 3, or 5 mg/kg) was orally administered 3 times: 0.5 h before and 1 h after occlusion, and 1 h after reperfusion (Figure 2A).

### 2.5. Neurological Deficit Score and Cerebral Infarct/Edema

Neurological deficit scoring of animals was evaluated at 24 h post-MCAO according to the Menzies method [22]. Motor coordination was assessed by a rotarod apparatus (775 Rotarod, IITC Life Science Inc., Woodland Hills, CA, USA). Animals pre-trained to remain on the revolving rod for 5 min were used for the surgical procedure, and the rotarod duration time was measured after MCAO/R. Data are shown for the longest time on the rotarod out of three trials. After the neurological deficit scoring and rotarod duration tests, the rats were sacrificed by decapitation under anesthesia and the brains were quickly collected. Then, each brain was cut into 2 mm-thick coronal sections in a rat brain matrix (ASI instruments Inc., Warren, MI, USA) and stained with 2% TTC solution for 15 min at 37 °C. The infarct and edema volumes for the brain tissue were measured by using ImagePartner software (SaramSoft Co., Ltd., Anyang, Korea). 

### 2.6. Biochemical Analysis

After 24 h of reperfusion, the ipsilateral rat brains were collected immediately after sacrifice and stored at −80 ℃. Homogenates were prepared fresh on ice using a 4-fold volume of 100 mM phosphate buffer (pH 7.4). Glutathione (GSH) levels in the ipsilateral brain were measured using a spectrophotometer (Sunrise, Tecan Austria GmbH, Grodig, Austria) as previously described [23]. The peroxidation of lipids in the ipsilateral brain was assessed by measuring the levels of thiobarbituric acid reactive substances (TBARS) at 532 nm (Sunrise, Tecan Austria GmbH, Grodig, Austria) as previously described [24].

### 2.7. Western Blotting

Total protein was extracted, and the protein concentration was determined by BCA assay. Approximately 35 μg of protein was loaded on to a 10% sodium dodecyl sulfate–polyacrylamide gel electrophoresis (SDS-PAGE) gel and then transferred to polyvinylidene fluoride (PVDF) membranes (Perkin Elmer Co., Waltham, MA, USA). The membranes were incubated with primary antibodies against Bcl-2, Bax, cleaved caspase-3, procaspase-3, i-NOS, COX-2, p-p38, p38, PI3K, p-Akt, Akt, p-CREB, CREB (1:1000), and β-actin (1:2000) followed by HRP-conjugated anti-rabbit or anti-mouse secondary antibodies (1:3000). Antibody binding was detected with an enhanced chemiluminescent reagent (Thermo Fisher Scientific Inc., Waltham, MA, USA), which is an HRP substrate. The images of target protein bands were captured using Lumino Graph 2 (ATTO Corporation, Tokyo, Japan) and luminescence intensity was quantified using Image J Fiji 1.53c analysis software (National Institutes of Health, Bethesda, MD, USA).

### 2.8. Statistical Analysis

Data are expressed as the mean ± standard error of mean (SEM) and analysed using a one-way analysis of variance followed by Tukey’s test using GraphPad Prism 5 (GraphPad Software, San Diego, CA, USA). *p*-values < 0.05 were considered significant.

## 3. Results

### 3.1. POP Attenuated Infarct/Edema Formation and Neurological Functional Deficits Induced by MCAO/R in Rats

As shown in Figure 2B, large ipsilateral cerebral infarctions were present in both the cortical and subcortical regions of the brain after MCAO/R. However, orally administered POP (at 1, 3, or 5 mg/kg) significantly decreased cerebral infarct and edema volumes induced by MCAO/R (Figure 2C,D). The body temperature of the rats was monitored for 6 h after cerebral reperfusion, and there was no significant difference in the body temperature between the groups (data not shown). Therefore, the neuroprotective effect of POP in the present study could not be related to a hypothermic effect.

Neurological scoring was assessed after 24 h of reperfusion using the Menzies system [22]. Sham-animals showed no neurological deficits (data not shown), while MCAO/R rats showed neurobehavioral deficits, such as circling movement and decreased grip strength of the contralateral forelimb. The increase in neurological functional deficit scores caused by MCAO/R was significantly reduced in the POP (at 1, 3, or 5 mg/kg)-treated groups compared to that of the vehicle-treated group (Figure 2E). Rotarod duration was assessed immediately after 24 h of reperfusion. MCAO/R rats showed hemiparesis and a loss of coordinated movements. Rotarod duration for rats in the MCAO/R group was markedly decreased compared to that of the sham group. The reduction in rotarod duration caused by MCAO/R was significantly alleviated by POP (at 1, 3, or 5 mg/kg) treatment (Figure 2F).

### 3.2. POP Alleviated GSH Depletion and Lipid Peroxidation after MCAO/R in Rat Brains

After MCAO/R, the GSH levels in the ipsilateral brains of vehicle-treated animals were significantly reduced compared to sham animals. However, POP (at 1, 3, or 5 mg/kg) significantly prevented the decrease in GSH levels caused by MCAO/R (Figure 3A). In addition, MCAO/R induced the increase of TBARS in the ipsilateral brain of vehicle animals compared to sham animals. However, POP (at 3 or 5 mg/kg) alleviated the increase of TBARS formation in the brain (Figure 3B).

### 3.3. POP Reversed Changes in the Expression of Apoptosis-Associated Proteins after MCAO/R in Ipsilateral Rat Brains

The expression ratio of Bcl-2/Bax markedly decreased in MCAO/R brains compared to that of the sham-operated group. However, POP administration (at 1, 3, or 5 mg/kg) significantly increased the expression ratio of Bcl-2/Bax (Figure 4A,B). MCAO/R also significantly increased the expression ratio of active caspase-3/procaspase-3 in the ipsilateral brain compared to that of the sham animals. However, POP (at 3 or 5 mg/kg) treatment significantly inhibited the MCAO/R-induced increase in the expression ratio of cleaved caspase-3/procaspase-3 (Figure 4A,C).

### 3.4. POP Reduced the Upregulation of p38 MAPK and Inflammation-Related Proteins after MCAO/R in Ipsilateral Rat Brains

The expression ratio of p-p38/p38 was increased in ipsilateral brains after MCAO/R, which was significantly inhibited by POP (at 1, 3, or 5 mg/kg) treatment (Figure 5A,B). MCAO/R induced over-expression of i-NOS and COX-2 in the ipsilateral brain compared to the sham-control group. However, treatment with POP (at 1, 3, or 5 mg/kg) reduced the upregulation of i-NOS and COX-2 in ischemic brains (Figure 5A,C,D).

### 3.5. POP Attenuated the Suppression of PI3K/Akt/CREB Signaling after MCAO/R in Ipsilateral Rat Brains

The expression levels of PI3K in the ipsilateral brain were lower in MCAO/R brains than in sham-operated animals. However, the reduced levels of PI3K expression in the ipsilateral brain caused by MCAO/R were upregulated in POP-treated (at 1, 3, or 5 mg/kg) rats (Figure 6A,B). Moreover, the expression ratio of p-Akt/Akt and p-CREB/CREB was decreased in ipsilateral brains after MCAO/R, which was significantly alleviated by POP treatment (at 1, 3, or 5 mg/kg) (Figure 6A,C,D).

## 4. Discussion

Ischemic stroke occurs when a blood vessel in the brain is obstructed by a blood clot or plaque [2]. The dramatic reduction in oxygen and nutrients at the ischemic core mediates metabolic failure, including adenosine triphosphate (ATP) depletion, membrane depolarization, excessive presence of excitatory amino acids, eventually resulting in irreversible neuronal damages within a few minutes [25]. Therefore, the early restoration of blood supply to the ischemic region may be the best option for minimizing neuronal damage. However, reperfusion by thrombolysis causes secondary neuronal damage by mediating complex signaling pathways for neuronal death and survival in the penumbra, the surrounding region of the ischemic core [26]. Since secondary damage in the penumbra that is induced by reperfusion is considered relatively salvageable, the development of neuroprotective agents targeting reperfusion injury may be a promising approach for the care of ischemic stroke. The rat MCAO/R model induced by the intraluminal suture method has been widely used as an animal model for ischemic stroke and is known to recapitulate aspects of human ischemia-reperfusion injury, including its neurobehavioral deficits, histological changes, and biochemical changes [27]. Similar to the progression of human ischemic stroke, MCAO/R induces primary neuronal damage in the subcortical region and secondary neuronal damage in the cortical region, which is correlated with neurological deficits in rats [27,28]. In the present study, POP reduced MCAO/R-mediated infarct/edema formation in rats, particularly in the cortical region, and alleviated the neurological deficits and the disturbances of motor coordination. These results indicate that the neuroprotective effect of POP may be partly related to the mechanisms involved in secondary brain injury in the penumbra.

Excessive increases in ROS caused by cerebral ischemia-reperfusion injury are involved in the early steps of cerebral damage after stroke [29,30]. The primary source of ROS under cerebral ischemia-reperfusion conditions is the mitochondrial electron transport chain, but other important sources, including xanthine oxidase, COX, lipoxygenase, and nicotinamide adenine dinucleotide phosphate oxidase, are also associated with the ROS outburst after reperfusion [31,32]. ROS reacts with DNA, proteins, and lipids, causing dysfunction of these molecules and neuronal apoptosis [33]. Since the brain is particularly vulnerable to lipid peroxidation due to the large amount of lipids and high oxygen dependence, the maintenance of the intrinsic anti-oxidants that scavenge free radicals is crucial for neuroprotection against ischemia-reperfusion injury [34]. GSH is one of the most abundant non-enzymatic anti-oxidants in the brain, and it protects neurons from oxidative damage by scavenging ROS [33,35]. In the current study, MCAO/R significantly reduced brain GSH content and increased lipid peroxidation, which might be associated with the upregulation of i-NOS and COX-2 in the brain. However, these effects were alleviated by treatment with POP, indicating that POP restored the free radical redox system and thereby prevented oxidative stress after cerebral ischemia-reperfusion.

Since mitochondria are both a primary origin and target for ROS, oxidative stress is also associated with the initiation of the intrinsic apoptotic pathway under ischemia-reperfusion conditions [32]. The Bcl-2 family members control mitochondrial integrity and determine the response to intrinsic apoptotic signals by regulating the release of mitochondrial proteins such as cytochrome c. Released factors interact with adaptor proteins to activate caspases, which trigger a biochemical cascade resulting in apoptosis [36,37]. In the present study, MCAO/R induced the downregulation of Bcl-2 and upregulation of Bax and active caspase-3 in the rat brain; these alterations in apoptosis-related proteins were significantly alleviated by treatment with POP. These results indicate that the neuroprotective effect of POP against MCAO/R-induced brain damage may be attributable to its anti-apoptotic potential, which contributes to the prevention of neuronal death, particularly in the penumbra.

The p38 MAPK signaling is associated with various pathways, including oxidative stress, apoptosis, excitotoxicity, and inflammation under ischemia-reperfusion conditions [7,8,9]. In particular, the p38 MAPK pathway has been reported as one of the key players in the inflammatory response after ischemic brain injury [9,11]. Upregulation of the p38 MAPK pathway results in overexpression of inflammatory mediators such as i-NOS and COX-2, which are also related to oxidative stress [14,15,38]. Oxidative stress mediated by i-NOS/COX-2 increases the production of pro-inflammatory cytokines, including IL-2, IL-6, and tumor necrosis factor-α through the upregulation of nuclear factor kappa-B signaling [39,40,41]. In this study, POP significantly suppressed MCAO/R-mediated upregulation of p-p38 MAPK expression in the brain, and the increased expression of i-NOS and COX-2 after MCAO/R was prevented by POP treatment. These results indicate that downregulation of p38 MAPK phosphorylation by POP might contribute to the reduction of ischemic brain damage by suppressing inflammation-mediated neuronal apoptosis.

The PI3K/Akt signaling plays a crucial role in the regulation of neuronal survival, proliferation, and inflammation after cerebral ischemia [42]. Activation of the PI3K/Akt pathway prevents ROS generation following the upregulation of the pro-apoptotic Bcl-2 family proteins, and eventually exerts neuronal survival after ischemic-reperfusion injury [16]. In addition, PI3K/Akt signaling has been reported to be also involved in post-ischemic axonal growth and regeneration through its downstream mediators, including endothelial NOS, glycogen synthase kinase-3 beta, and CREB [43,44,45]. In particular, the phosphorylation of CREB following activation of the PI3K/Akt pathway induces post-stroke restoration through the upregulation of neurogenic agents, such as nerve growth factor and brain-derived neurotrophic factor [45,46]. In the present study, POP prevented the downregulation of PI3K/Akt/CREB signaling after MCAO/R, which may be beneficial for neuronal survival after ischemic-reperfusion injury. The involvement of POP in the PI3K/Akt/CREB pathway suggests it may have potential as a neuro-rehabilitation agent, so further studies are needed.

Dietary TAGs are one of the fundamental lipid components in the human diet and function as carriers of energy and essential fatty acids in the body [47]. Although high TAG levels in the serum are associated with pathophysiological processes such as thrombus formation and endothelial dysfunction that can lead to a higher risk of ischemic stroke, the significance of dietary TAGs to ischemic stroke remains controversial [48,49,50]. The structure and esterification of the fatty acids contribute to the physiological properties of dietary TAGs, including absorption, metabolism, and distribution [51]. In plant-derived TAGs, saturated fatty acids are predominantly located at the external positions *sn*-1 and *sn*-3, and unsaturated fatty acids occupy the internal position *sn*-2. Due to preferential hydrolysis of fatty acids at the *sn*-1 and *sn*-3 positions by pancreatic lipase, an unsaturated fatty acid at the *sn*-2 position in plant-derived TAGs is more advantageous for absorption as a 2-monoacylglycerol (MAG) [51]. Therefore, it is considered that oleic acid at the *sn*-2 position of POP derived from RBO could be relatively well absorbed as a MAG in the intestine. Furthermore, the neuroprotective effects of oleic acid against ischemic damage in animal models have been demonstrated [52], suggesting a possible contribution from oleic acid at the *sn*-2 position of POP in the neuroprotective effects of POP in the current study.

## 5. Conclusions

In conclusion, the present study has demonstrated the neuroprotective effects of POP against MCAO/R-induced cerebral ischemic damage in rats. Our mechanistic studies have shown that the neuroprotective effects of POP are involved in p38 MAPK inhibition and PI3K/Akt/CREB pathway activation, which is associated with anti-oxidant, anti-apoptotic, and anti-inflammatory effects. These results suggest that POP possesses multi-targeted neuroprotective effects, and this pleiotropic potential makes POP a promising agent for managing cerebral ischemia-reperfusion injury-related disorders. 

## Figures and Tables

**Figure 1 nutrients-14-01380-f001:**
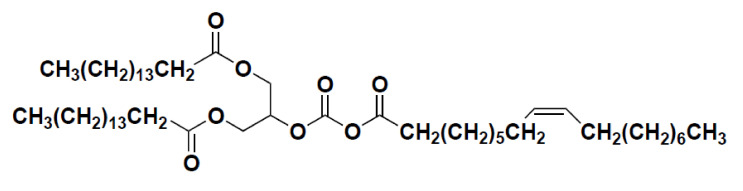
Chemical structure of POP. POP, 1,3-dipalmitoyl-2-oleoylglycerol.

**Figure 2 nutrients-14-01380-f002:**
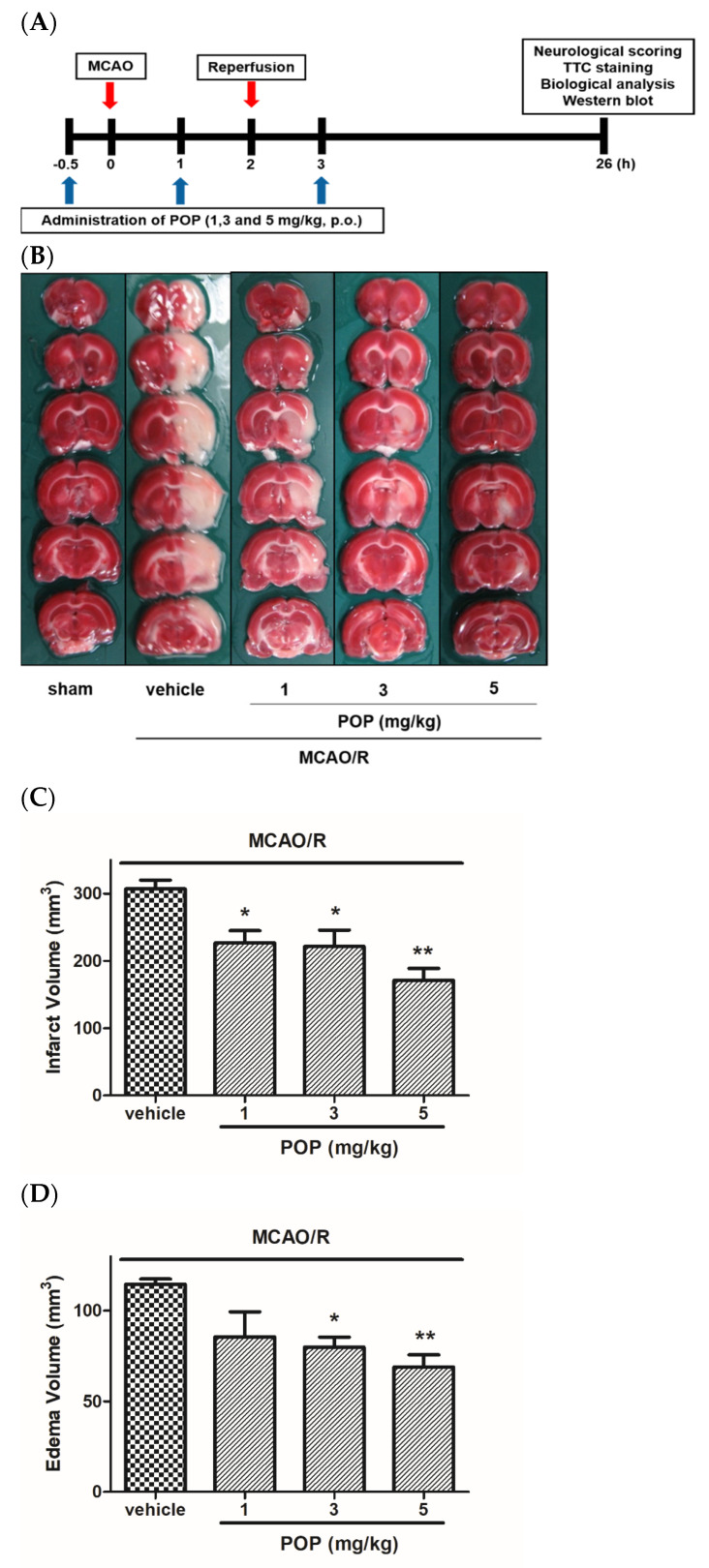
Effects of POP on neurological deficits and brain infarct/edema formation after the MCAO/R in rats. (**A**) Schematic of the in vivo experiment. (**B**) Representative photographs of TTC staining of brain slices (2 mm) showing the infarct area. Quantitative analysis of (**C**) the infarct volume and (**D**) the edema volume in the ipsilateral hemisphere. Quantitative analysis of (**E**) the neurological score and (**F**) rotarod duration. Values are expressed as the mean ± SEM (*n* = 6−7). ## *p* < 0.01 versus sham; * *p* < 0.05 and ** *p* < 0.01 versus the vehicle-treated group (Tukey’s test). MCAO/R, middle cerebral artery occlusion/reperfusion; TTC, 2,3,5-triphenyl tetrazolium chloride; SEM, standard error of mean, p.o, per oral, h, hours.

**Figure 3 nutrients-14-01380-f003:**
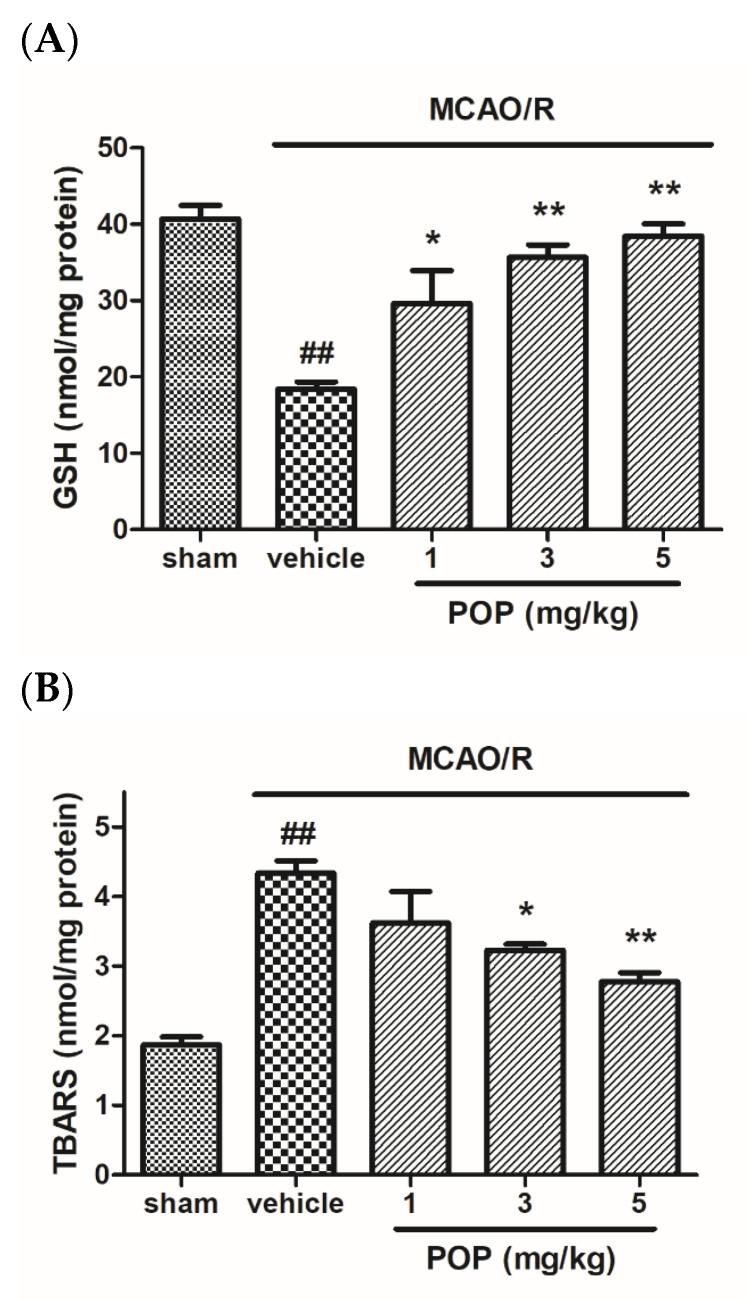
Effects of POP on GSH and TBARS levels in ipsilateral rat brains after MCAO/R. Quantification of (**A**) GSH and (**B**) TBARS levels in ipsilateral rat brains 24 h after MCAO/R. Values are expressed as the mean ± SEM. ## *p* < 0.01 versus sham; * *p* < 0.05 and ** *p* < 0.01 versus the vehicle-treated group (Tukey’s test). GSH, glutathione; TBARS, thiobarbituric acid reactive substances.

**Figure 4 nutrients-14-01380-f004:**
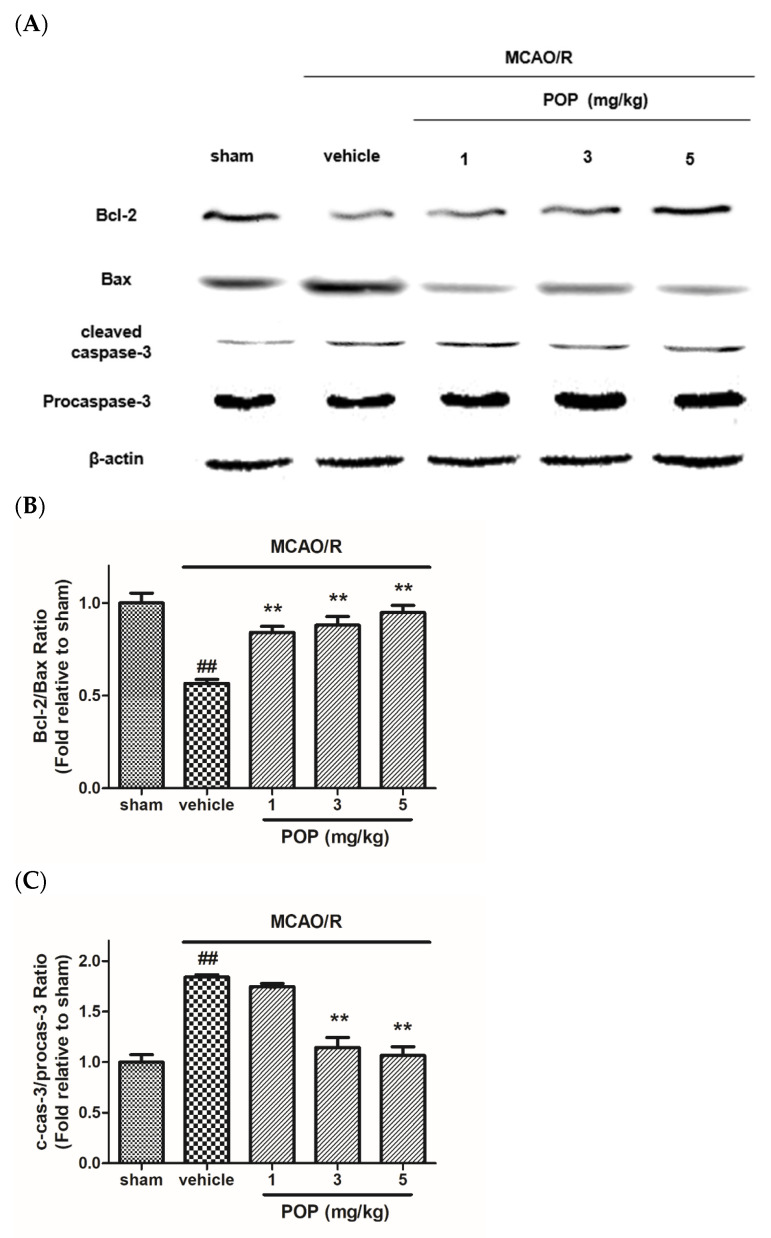
Effects of POP on the expression of apoptosis-associated proteins in ipsilateral rat brains after MCAO/R. (**A**) Representative Western blot images of proteins from ipsilateral brains 24 h after MCAO/R. Quantification of the relative ratio of (**B**) Bcl-2/Bax and (**C**) cleaved caspase-3/procaspase-3 expression versus the sham control. Values are expressed as the mean ± SEM. ## *p* < 0.01 versus sham; ** *p* < 0.01 versus the vehicle-treated group (Tukey’s test). Bcl-2, B-cell lymphoma-2; Bax, B-Bcl-2 associated X protein; c-cas-3, cleaved caspase-3.

**Figure 5 nutrients-14-01380-f005:**
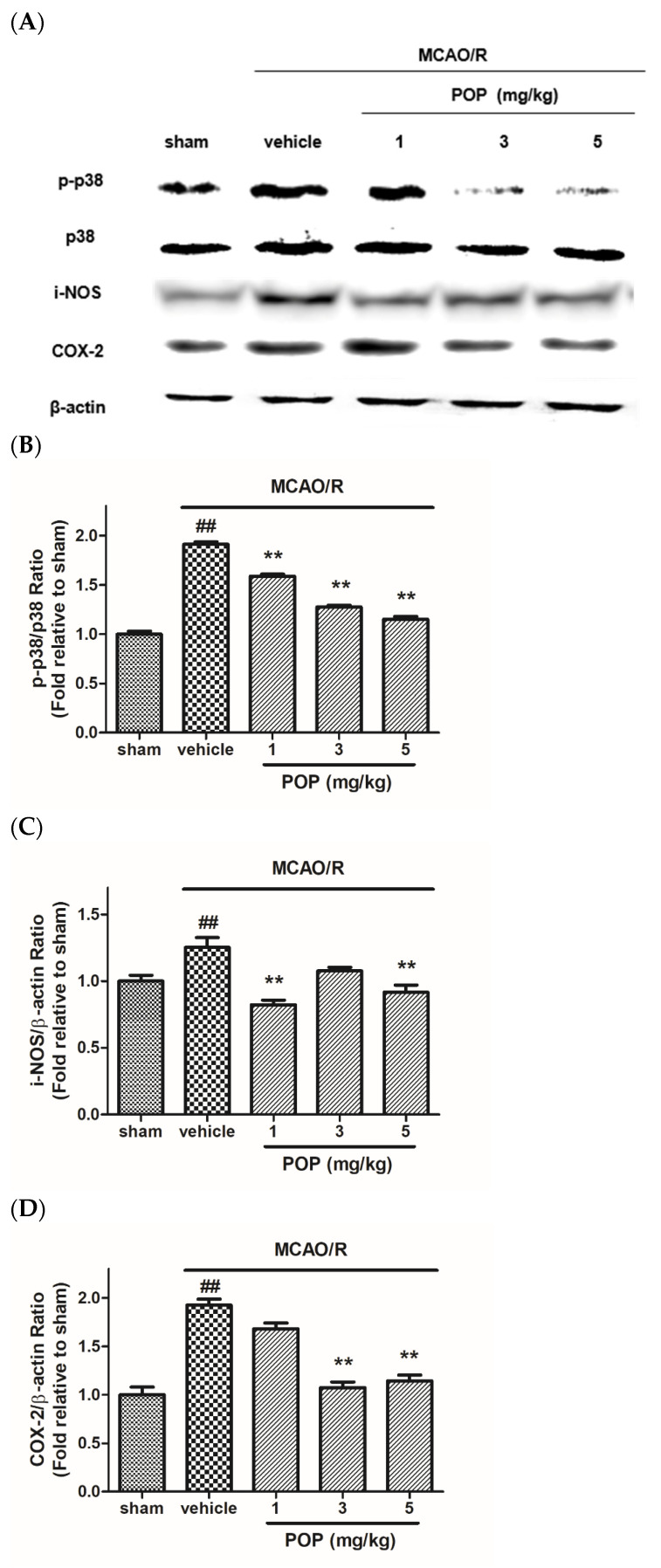
Effects of POP on the expression of p38, i-NOS, and COX-2 in ipsilateral rat brains after MCAO/R. (**A**) Representative Western blot images of proteins from ipsilateral brains 24 h after MCAO/R. Quantification of the relative ratio of (**B**) p-p38/p-38, (**C**) i-NOS/β-actin, and (**D**) COX-2/β-actin versus the sham control. Values are expressed as the mean ± SEM. ## *p* < 0.01 versus sham; ** *p* < 0.01 versus the vehicle-treated group (Tukey’s test). p38, signaling transduction pathway; p-p38, phospho-p38; i-NOS, inducible nitric oxide synthase; COX-2, cyclooxygenase-2.

**Figure 6 nutrients-14-01380-f006:**
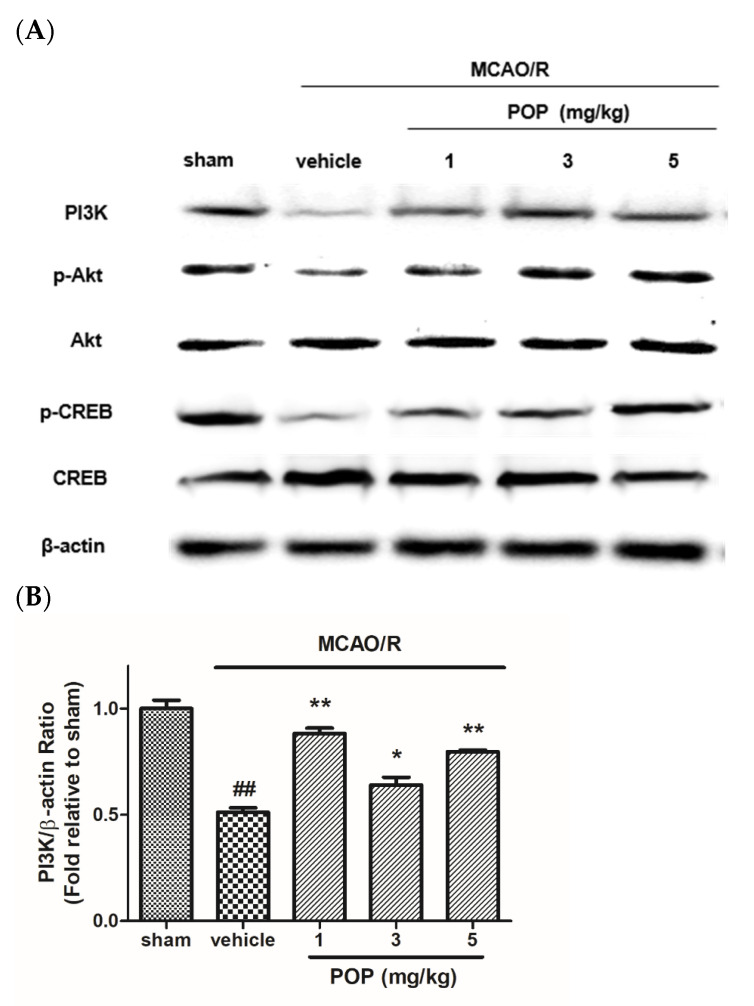
Effects of POP on the expression of PI3K, Akt, and CREB in ipsilateral rat brains after MCAO/R. (**A**) Representative Western blot images of proteins from ipsilateral brains 24 h after MCAO/R. Quantification of the relative ratio of (**B**) PI3K/β-actin, (**C**) p-Akt/Akt, and (**D**) p-CREB/CREB versus the sham control. Values are expressed as the mean ± SEM. ## *p* < 0.01 versus sham; * *p* < 0.05 and ** *p* < 0.01 versus the vehicle-treated group (Tukey’s test). PI3K, phosphatidylinositol 3′-kinase; p-Akt, phosphorylated protein kinase B, Akt, protein kinase B; p-CREB, phosphorylated cyclic (adenosine monophosphate) AMP responsive element-binding protein; CREB, cyclic AMP responsive element-binding protein.

## Data Availability

The data will be made available from the authors upon reasonable request.

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
