# Peer review of "Neuroprotective Effect of 1,3-dipalmitoyl-2-oleoylglycerol Derived from Rice Bran Oil against Cerebral Ischemia-Reperfusion Injury in Rats"

_nutrients, 2022, doi:10.3390/nu14071380_

Round 1
Reviewer 1 Report
This study is one of the novel study which investigate the neuroprotective effect of 1,3-dipalmitoyl-2-oleoylglycerol derived from rice bran oil against cerebral ischemia-reperfusion injury in rats. It provides a valuable data about the effect of 1,3-dipalmitoyl-2-oleoylglycerol as anti-oxidant, anti-apoptotic, and anti-inflammatory action.
Also, this study exhibited that the POP might be effectively applied for the management of cerebral ischemic injury-related diseases.
- This study answered the research question clearly.
- Authors have to acknowledge the support from the institutions or research centers that effectively support and provide a facility for doing this research.
- References should be standardized according to the instruction of the journal. full stop should be added for the abbreviated journal names.
- Some tying errors are there, it needs correction.
Author Response
- Response to reviewer 1
- Authors have to acknowledge the support from the institutions or research centers that effectively support and provide a facility for doing this research.
answer; We added it.
- References should be standardized according to the instruction of the journal. full stop should be added for the abbreviated journal names.
answer; We modified it.
- Some typing errors are there, it needs correction.
answer; We modified it.

Reviewer 2 Report
The Authors focused on a study of the Neuroprotective Effect of 1,3-dipalmitoyl-2-oleoylglycerol Derived from Rice Bran Oil Against Cerebral Ischemia-Reperfusion Injury in Rats. This is an interesting and comprehensive study. The article is well structured.
In my opinion:
- The abstract presents an accurate description of this review.
- An Authors was conducted adequate literature review.
- The references support the rationale for reporting the study.
- The subjects are described adequately.
- The management of the study is effectively described.
- Valid and reliable outcome measures are utilized.
- The management of the manuscript is effectively described.
- The conclusions are appropriate.
Overall impression about the quality of the study is good.
Line 12 – “1, 3-dipalmitoyl-2-oleoylglycerol”, please fix it to “1,3-dipalmitoyl-2-oleoylglycerol”
Line 12 – triacylglyceride, please add abbreviation to name (first time in the manuscript)
Line 29 – “1; 3-dipalmitoyl-2-oleoylglycerol”, please fix it to “1,3-dipalmitoyl-2-oleoylglycerol”
Line 61 – Triacylglyceride (TAG) - it is not necessary to explain the abbreviations, it was before, use just shorts
Line 85, 86 – different font, please fix it
Author Response
Response to Reviewer 2
Line 12 – “1, 3-dipalmitoyl-2-oleoylglycerol”, please fix it to “1,3-dipalmitoyl-2-oleoylglycerol”
answer; We modified it.
Line 12 – triacylglyceride, please add abbreviation to name (first time in the manuscript)
answer; Since triacylglyceride appears only once in abstract, it is considered unnecessary to use the abbreviation in abstract.
Line 29 – “1; 3-dipalmitoyl-2-oleoylglycerol”, please fix it to “1,3-dipalmitoyl-2-oleoylglycerol”
answer; We modified it.
Line 61 – Triacylglyceride (TAG) - it is not necessary to explain the abbreviations, it was before, use just shorts
answer; Since Triacylglyceride (TAG) appears for the first time in the main text, an abbreviation is indicated.
Line 85, 86 – different font, please fix it
answer; We modified it.
